# A modelling assessment of the impact of control measures on highly pathogenic avian influenza transmission in poultry in Great Britain

Christopher N. Davis[1,2,3]*, Edward M. Hill[4,5], Chris P. Jewell[6], Kristyna Rysava[1,2,3,7], Robin N. Thompson[7], Michael J. Tildesley[1,2,3]

**1** Zeeman Institute for Systems Biology and Infectious Disease Epidemiology Research (SBIDER), University of Warwick, Coventry, United Kingdom, **2** Mathematics Institute, University of Warwick, Coventry, United Kingdom, **3** School of Life Sciences, University of Warwick, Coventry, United Kingdom, **4** Civic Health Innovation Labs and Institute of Population Health, University of Liverpool, Liverpool, United Kingdom, **5** NIHR Health Protection Research Unit in Emerging and Zoonotic Infections, University of Liverpool, Liverpool, United Kingdom, **6** Department of Mathematics and Statistics, Lancaster University, Lancaster, United Kingdom, **7** Mathematical Institute, University of Oxford, Oxford, United Kingdom

* c.davis.7@warwick.ac.uk

## Abstract

Since 2020, large-scale outbreaks of highly pathogenic avian influenza (HPAI) H5N1 in Great Britain have resulted in substantial poultry mortality and economic losses. Alongside the costs, the risk of circulation leading to a viral reassortment that causes zoonotic spillover raises additional concerns. However, the precise mechanisms driving transmission between poultry premises and the impact of potential control measures in Great Britain, such as vaccination, are not fully understood. We have developed a spatial transmission model for the spread of HPAI in poultry premises calibrated to infected premises data for the 2022–23 season using Markov chain Monte Carlo. Our results indicate that reducing the susceptibility of the premises surrounding an identified infected premises (for example, through enhanced biosecurity measures and/or vaccination) can substantially reduce the overall number of infected premises. Our findings highlight that enhanced control measures could limit the future impact of HPAI on the poultry industry and reduce the risk of broader health threats.

## Author summary

Highly pathogenic avian influenza is an infectious disease that has caused a substantial number of outbreaks in bird populations around the world. This includes poultry populations in Great Britain, where the largest number of infected poultry premises were reported in the 2022–23 season. This identifies the need for mathematical models to be calibrated to outbreak data, such that we can have a greater understanding of the disease transmission process and the potential measures that could be used to avert future infections. We have developed a model to describe temporal changes in avian influenza infections in poultry premises in

**Data availability statement:** We have used data on HPAI cases in poultry (consisting of the location of poultry premises, number of kept birds and report date for infection) and Great British poultry premises demography (consisting of the centroid of poultry premises locations and number of kept birds). The data is sensitive and can only be provided on a case-by-case basis to academic researchers upon submission of a research proposal. We are not in a position to directly share these data as this would breach GDPR—the premises level data contain potentially identifiable information. To access the data, contact the Animal and Plant Health Agency (APHA) (enquiries@apha.gov.uk). The model code and scripts needed to generate the main figures can be found at https://github.com/cnd27/HPAI_control_measures/.

**Funding:** This work was supported by an Ecology and Evolution of Infectious Diseases joint National Science Foundation (NSF) and Biotechnology and Biological Sciences Research Council (BBSRC) grant (BB/X005224/1 to MJT and CND). CND had full salary support from the BBSRC and MJT had partial salary support from BBSRC. The work was also supported by a BBSRC grant (BB/X016137/1 to MJT, RNT, EMH and KR). KR had full salary support from the BBSRC and MJT, RNT and EMH had partial salary support from BBSRC. EMH is affiliated to the NIHR Health Protection Research Unit in Emerging and Zoonotic Infections (NIHR207393). EMH is funded by The Pandemic Institute, formed of seven founding partners: The University of Liverpool, Liverpool School of Tropical Medicine, Liverpool John Moores University, Liverpool City Council, Liverpool City Region Combined Authority, Liverpool University Hospital Foundation Trust, and Knowledge Quarter Liverpool (EMH is based at The University of Liverpool). The funders had no role in

Great Britain over the course of an epidemic season. We then simulate reactive enhanced control strategies that reduce susceptibility of poultry to show that there is a large benefit in localised interventions around an infected poultry premises to reduce further transmission. These results underscore the need for effective control strategies to limit the continued circulation of avian influenza and the threat to public health.

## Introduction

Highly pathogenic avian influenza (HPAI) poses a substantial ongoing threat to the poultry industry. Since 2020, the emergence of a reassorted genotype of H5N1 viruses within clade 2.3.4.4b has been associated with widespread outbreaks of HPAI in both wild birds and poultry worldwide [1]. The currently circulating H5N1 clade 2.3.4.4b strain has a large fitness advantage over previously circulating viruses [2]. It has impacted a larger number of bird species, notably causing large mortality events in seabird colonies [3,4]. Moreover, spillover into mammals, including evidence of mammal-to-mammal transmission, such as in cattle in the USA [5,6], as well as an increasing number of confirmed human infections [7,8], indicate the zoonotic potential of the virus and the potential risk of a future pandemic occurring [9].

In Great Britain, there have been annual epizootic events affecting poultry premises since 2020, with approximately 200 infected premises (IPs) during the 2022–23 season (1 October 2022 — 30 September 2023) [10]. This has resulted in the culling of millions of birds to prevent further spread of infection, at substantial economic cost [2,11]. The outbreaks traditionally followed a seasonal pattern, driven by the arrival of infection from migratory wild birds in the autumn and winter months, with very few infections in poultry during the summer [12]. However, with a broader range of wild bird host species infected and endemic circulation in these resident wild birds [13], while the seasonal pattern remains, there has been an increased incidence in observed IPs over the summer months, with HPAI detections throughout the summers since 2022 [10].

Most reported IPs were likely associated with spillover from local wild bird populations in 2022–23 [14], while infected migratory birds commonly increase the geographical spread of infection [15]. The majority of transmission to poultry has generally been attributed to Charadriiformes (such as waders, gulls and auks) and Anseriformes (such as ducks and geese) and in most cases this is due to direct or indirect contact, or through contamination of bedding or feed [14]. Within poultry premises, chickens and turkeys infected with HPAI typically show more severe symptoms or have higher mortality compared to ducks and geese [16], although the latter may have similar levels of viral shedding without the symptoms or mortality. This difference, however, may be less obvious for some genotypes of the circulating H5N1 clade 2.3.4.4b, as they are particularly well adapted to ducks [17–19].

study design, data collection and analysis, decision to publish, or preparation of the manuscript.

**Competing interests:** I have read the journal's policy and the authors of this manuscript have the following competing interests: EMH is an Editorial Board member for PLOS Computational Biology.

Premises-to-premises transmission has been reported in Europe [20], but there is little evidence that this is common in Great Britain. Phylogenetic analyses have identified premises-to-premises transmission as being likely for only a few select IPs during 2020–2022 [21]. Where premises-to-premises transmission does occur, it is likely due to the movement of vehicles, shared equipment or personnel between premises, or by the transport of infected birds to a new premises [14,22,23]. During the 2022–23 season, airborne transmission between premises was unlikely since evidence suggests that airborne particles containing HPAI virus can only travel very short distances (up to 10 metres) [24]. With that understanding of the likely modes of transmission, biosecurity measures are therefore essential to prevent the introduction of the HPAI virus into premises. Biosecurity measures can include increased disinfection and cleaning, management and treatment of water, the prevention of wild-bird access to housing and feed storage, changing of footwear for poultry workers moving between premises, and improved fencing to reduce contact between poultry and wild bird species [25,26]. In Great Britain, Avian Influenza Prevention Zones (AIPZ) have been introduced that legally require poultry owners to follow strict biosecurity measures and can include mandatory housing orders [27,28]. As of 2025, the use of HPAI vaccines for poultry is not permitted in Great Britain [29].

In the 2022–23 season, there was a national AIPZ in place from 17 October 2022 to 4 July 2023 [27], with a regional AIPZ in the counties of Norfolk, Suffolk, and parts of Essex before this from 27 September 2022, and so covering almost all the infections on premises seen during the season. A national housing order was also in place from 7 November 2022 to 18 April 2023 [28], with this beginning regionally in the East of England from 12 October 2022. Additional 3 km Protection Zones were established around IPs to limit transmission, alongside 10 km (7 km in addition to the 3 km) Surveillance Zones with increased record keeping and monitoring for infections [30]. For our mathematical analysis, we choose to model the 2022–23 season.

Mathematical models for HPAI outbreaks amongst poultry premises have been used to estimate the probability that large outbreaks occur in a variety of geographical settings and to identify areas at high risk of infection. In the context of Great Britain, spatial models have historically helped to determine the probability of outbreak clusters [31,32]. Spatial modelling studies have also identified key parameter values for model simulations in other settings, such as Bangladesh [33] and Thailand [34]. Modelling approaches have been developed to produce risk maps of HPAI H5N1 Clade 2.3.4.4b spillover in Europe [35] and the USA [36]. Other studies have sought to identify suitable control policies for HPAI, including vaccination, ring culling, increased surveillance, and contact tracing in countries such as Vietnam [37], Bangladesh [38], South Korea [39], and France and the Netherlands [40]. However, no known studies have been used to infer the transmission dynamics of recent outbreaks of HPAI in Great Britain, or to assess the impact of potential control strategies.

In this manuscript, we extended the work of previous modelling approaches, such as Jewell et al. [31] and Hill et al. [33], to adapt a spatial individual poultry premises-based model for Great Britain. We used this model to capture the infection dynamics

in poultry premises across the 2022–23 season, the season with the largest number of HPAI-infected poultry premises (at the time of writing in 2025). We believe this is the first mechanistic model fitted to recent HPAI epidemic data in Great Britain. We used Markov chain Monte Carlo (MCMC) to parameterise the model using notification data from the 2022–2023 season, evaluating the quality of the model fit with model simulations and fitting statistics. We then considered the impact of biosecurity measures or the potential use of vaccines by implementing enhanced control measures in the immediate area surrounding an IP. Using model simulations, we explored how varying the stringency, duration, and area of the enhanced control zone, which reduces poultry susceptibility, affects the number of reported IPs. These results quantify the impact of additional control measures for a range of possible scenarios.

## Methods

### Data

We obtained demographic data on poultry premises in Great Britain and poultry case data from the Animal and Plant Health Agency. The demographic data include the centroid of each premises polygon (defined as a CPH — County/Parish/Holding number — entity), as well as the number of poultry reported as kept by species. We use demographic data that were registered on 1 December 2022, which falls within our fitting period. To provide the required inputs into the spatial model, we aggregated these flock counts into three categories for each premises: Galliformes (chickens, turkeys, etc.), waterfowl (ducks, geese, etc.), and other birds. We chose these categories to limit the number of bird types considered for a reasonable number of parameters in the model while allowing for differences in the transmission and susceptibility characteristics of different species. There were 48660 premises within our data set. While we assumed this was a complete list of all poultry in the country, some premises of any size may be missing from our data set and, in particular, we noted that it was not a legal requirement for premises with fewer than 50 captive birds to officially register their birds in the considered time period (before 1 October 2024), indicating that some small premises may be missing [41].

In this study, we considered premises with H5N1-infected birds within the 2022–23 epidemiological season: 1 October 2022 – 30 September 2023. This period contains a peak in infections in the autumn/winter, with relatively fewer cases in the summer months, consistent with the known seasonality in infections [12]. There were 200 premises with confirmed H5N1 infection in the 2022–23 season (Fig 1).

At any point in time, we assume that all IPs are included in the data set (due to identification by poultry exhibiting clinical signs or suffering mortality), unless they have yet to be reported. The IPs will be reported because HPAI is a notifiable disease in the UK, and poultry owners are legally obliged to report suspected infections to the Animal and Plant Health Agency (APHA) [42]. Upon reporting or 'notification', a veterinary inspector typically visits the premises to test for the presence of HPAI within the poultry, where, on disease confirmation, all susceptible poultry will be culled unless specific exemption criteria apply (such as some birds from zoos, circuses or pet shops) [43,44]. Additional measures such as movement bans, disinfection, and other restrictions may also be put in place.

We link data on IPs to the poultry premises register data by matching the coordinates of the premises with reported infections to the closest premises with similar reported poultry numbers. Full details are provided in S1 Text. This means we have data on all premises contained in the poultry keeper register data for Great Britain on 1 December 2022, and for those premises where H5N1 infection was detected, the dates that notification occurred before all birds were culled on the premises.

### Model

Our model is formulated as a discrete-time, individual-based, spatially explicit compartmental model for individual poultry premises. Our model conceptualisation uses a similar structure to models presented in previous studies on HPAI and other livestock diseases [33,45–47]. At a given time, each premises can be in one of five given states: susceptible to HPAI

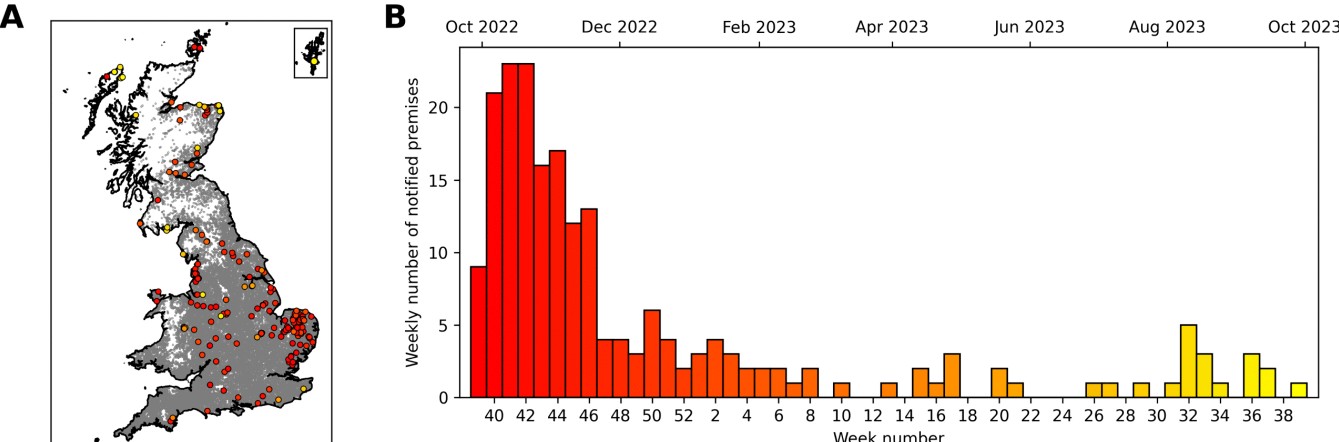

**Fig 1. Reported cases of H5N1 in poultry premises in Great Britain for the 2022–23 season, 1 October 2022 – 30 September 2023.** (A) A map of all poultry premises in the poultry keeper register on 1 December 2022 in Great Britain (grey dots), and the locations of infected premises (IPs) within this time period (coloured dots). The colour of the IP dots indicates the time within the season that the notification occurred, with red dots at the start of the season and yellow dots at the end. The inset map shows the Shetland Islands to the north of mainland Scotland. Source of map boundaries: Office for National Statistics licensed under the Open Government Licence v.3.0. The shapefiles used can be found at https://geoportal.statistics.gov.uk/datasets/5a393192a58a4e50baf87eb4d64ca828_0/explore. (B) A bar chart of the number of notified poultry premises in each week of the 2022–23 season. The colour scale of the bars indicates the progression of time, as shown in (A). The secondary axis at the top shows the start of selected months in this time period.

infection S, exposed to HPAI infection (i.e., infected but not yet able to transmit infection) E, infectious and able to transmit infection I, notified as infected but still infectious N, and removed by culling R. For each premises $i$, we denote $E_i$ as the time of infection and so when the premises becomes exposed. This similarly applies to the time of the onset of infectiousness ($I_i$), the time of notification ($N_i$), and the time of culling ($R_i$). We assume the infection events occurred according to the given infection rate $\lambda_j(t)$.

The infectious pressure on premises $j$ is given by:

$$\lambda_j(t) = \epsilon(t) + \sum_{i \in \mathcal{I}(t)} \beta_{ij} + \gamma_1 \sum_{i \in \mathcal{N}(t)} \beta_{ij}, \tag{1}$$

where $\mathcal{I}(t)$ and $\mathcal{N}(t)$ are the sets of premises that are infected and notified at time $t$, respectively. This describes two components of infection: (i) a time-varying term, $\epsilon(t)$, for the background infection directly caused by spillover from wild bird populations; (ii) local infections in regions close to other poultry premises that are currently infectious (infectious I or notified N) [48], captured by the two summation terms in Eq 1. The local force of infection from premises $i$ to premises $j$ is given by $\beta_{ij}$, and $\gamma_1$ is a scaling of the force of infection for premises that have been notified as infectious, as opposed to infectious but not yet notified. The local infection component from other poultry premises could reflect a range of different transmission modes, for example: direct transmission by infected poultry; indirect transmission via wild birds as bridging vectors, or; transmission of virus on shared farming equipment or by staff. We also note that cases on premises in close proximity could be indicative of an increased presence of H5N1 in local wild bird populations, causing multiple spillover events. Therefore, both components of the infectious pressure may be due to wild bird spillover, with known poultry infections spatially indicating potential higher-risk areas. No transmission routes are excluded in these terms, but likewise none are explicitly modelled, unlike, for example, in models in which networks of vehicle movements were modelled explicitly.

The constituent terms of $\lambda_j(t)$ are further broken down to describe their functionality within the model. The background infection term:

$$\epsilon(t) = \epsilon_0 \exp\left(-\nu_0 \left(1 + \cos\left(2\pi\left(\frac{t}{365} - \nu_1\right)\right)\right)\right),\tag{2}$$

shows seasonal variation in its influence peaking on day $\nu_1$ in each year [12]. The force of infection on premises $j$ by premises $i$ is given by:

$$\beta_{ij} = \gamma_0\left(\left(\frac{x_{0i}}{\bar{x}_0}\right)^{\psi_0} + \xi_1\left(\frac{x_{1i}}{\bar{x}_1}\right)^{\psi_1} + \xi_2\left(\frac{x_{2i}}{\bar{x}_2}\right)^{\psi_2}\right)\left(\left(\frac{x_{0j}}{\bar{x}_0}\right)^{\phi_0} + \zeta_1\left(\frac{x_{1j}}{\bar{x}_1}\right)^{\phi_1} + \zeta_2\left(\frac{x_{2j}}{\bar{x}_2}\right)^{\phi_2}\right)K_{ij}.\tag{3}$$

The force of infection $\beta_{ij}$ therefore consists of four components. A multiplicative factor $\gamma_0$; the infectivity of the premises $i$, which is the sum of the number of birds of the three species types ($x_{ki}$, $k = 0, 1, 2$ for the three species types) divided by the mean value of the number of birds of each type across all premises ($\bar{x}_k$) raised to the power of the infectivity exponent ($\psi_k$) and multiplied by infectivity factor ($\xi_k$, where $\xi_0 = 1$); the susceptibility of premises $j$, which takes a similar form to the infectivity, but with susceptibility exponents ($\phi_k$) and factors($\zeta_k$); and the transmission kernel $K_{ij}$, which is a function of distance between the infected and susceptible premises.

We include exponents on the number of birds in the infectivity and susceptibility components because previous studies have found that non-linear terms provide a better fit to epidemic data [49]. In this study, we also assume that the kernel takes a Cauchy form with:

$$K_{ij} = K(d_{ij}) = \frac{\delta}{\left(\delta^2 + d_{ij}^2\right)^\omega}.\tag{4}$$

Therefore, the force of infection $\lambda_j(t)$ is only spatially dependent on the distance to other infectious and notified premises. We also consider an exponential kernel in Fig G in S1 Text to consider how the shape of the kernel impacts our results.

Given a timestep $[t, t + \delta t)$ where $\delta t = 1$ day, the probability that a susceptible premises becomes exposed on a given day is given by:

$$p_j(t) = 1 - \exp\left(-\lambda_j(t)\delta t\right),\tag{5}$$

and from this exposure event, we then assume the infection progresses in discrete time through the infection states after a specified number of days. We fix the latent period and time to culling from notification for improved model tractability, but leave the time to notification as variable for each premises in the fitting process, since we expect this will have the most variance in reality due to differences in premises-level surveillance and visibility of symptoms in birds.

In further detail, we fix the time spent in the exposed class as four days before moving to the infectious class, since the incubation period is generally less than seven days [50]. Four days is in agreement with the between-flock latency period of previous studies [51]. The time in the infectious class before notification is fitted for each premises, using a prior of $N_j - I_j \sim \text{Gamma}(4, 2)$ for the number of days to notification. Gamma$(a, b)$ describes the Gamma probability distribution with shape $a$ and scale $b$. This allows for individual differences dependent on the specific premises and provides an estimate that falls within the typical distribution (noting the mean value of our prior is eight days) [52]. The time from notification to culling the birds is taken as a fixed three days, which is consistent with the average report and confirmation dates in the data set.

## Model parameters and fitting

The model parameter values are determined using a Bayesian inference framework, where the parameters are updated using adaptive Metropolis–Hastings in a Markov chain Monte Carlo (MCMC) method [53,54]. This method has been implemented similarly in several studies [31,46,47,55].

We fit sixteen model parameters with one fixed parameter $\omega = 1.3$ to give the shape of our transmission kernel, based on previous studies [47]. More details are given in S1 Text. See Table 1 for full details on all the model parameters. Prior distributions for these parameters are chosen such as to provide variance around a mean value elicited from expert opinions, or to be uninformed.

Also included within the fitting algorithm is a reversible-jump update, which allows the addition (and removal) of undetected or 'occult' infections within the model to assess how their presence changes the likelihood [46]. In this step, which occurs after each update to the parameter values, a fixed number of additional model alterations are considered: the change of the time to notification for a given premises, the addition of a new IP as an 'occult' infection that is yet to be notified (and so appear in the data set), and the removal of any previously added 'occult' infections.

## Model projections and enhanced control strategies

The fitted model can be used to stochastically simulate the epidemic from a given set of initially infected premises to compare back to the observed data. For the 2022–23 season, our initial conditions are a fixed set of 31 infected (exposed, infectious, or notified) premises in all simulations across East of England (23 IPs), West Midlands (2), North West (2), East Midlands (1), South West (1), South East (1) and Scotland (1). The numbers were determined from our data set, using the known notification times, where the infection status will vary between simulations, given the distributions for the time to notification. The simulations are performed using a tau-leaping algorithm [56] in the grid-based conditional subsample algorithm for computational speed [57,58]. Using the conditional subsample algorithm, we divided the country into 10 km grid cells to first assess whether any transmission events occurred to any premises within each grid cell. We then considered pairwise transmission to premises within that cell. This approach omits many unlikely calculations for transmission over large distances. The algorithm is fully described in Sellman et al [58].

Additionally, we perform counterfactual simulations based on alternative scenarios using the same methodology. These model simulations specifically investigate how enhanced control might be enacted in response to an IP. We assume a baseline level of biosecurity, which is determined by the fitting process for the 2022–23 season, where, in reality, an AIPZ and housing order were in place for the majority of the season. Enhanced control measures could include increased

**Table 1**. **List of model parameters with descriptions.** Prior distributions are given for fixed parameters where Gamma$(a, b)$ is a gamma distribution with shape $a$ and scale $b$ and Beta$(a, b)$ is a beta distribution with shape parameters $a$ and $b$. Note that $\xi_0 = 1$ and $\zeta_0 = 1$ are not listed as $\xi_k$ and $\zeta_k$ are relative to these parameters, respectively, for $k = 1, 2$.

| Parameters | Description | Prior/Value |
|---|---|---|
| $\epsilon_0$ | Baseline infectious pressure | Gamma$(1, 1 \times 10^{-5})$ |
| $\gamma_0$ | Infectious pressure contribution from IPs | Gamma$(1, 0.01)$ |
| $\gamma_1$ | Multiplicative factor for infectious pressure contribution for notified premises | Gamma$(1, 0.8)$ |
| $\delta$ | Decay of transmission rate between premises in the transmission kernel | Gamma$(2, 1)$ |
| $[\psi_0, \psi_1, \psi_2]$ | Exponential terms in infectious pressure for infectious premises | Beta$(2, 2)$ |
| $[\phi_0, \phi_1, \phi_2]$ | Exponential terms in infectious pressure for susceptible premises | Beta$(2, 2)$ |
| $[\xi_1, \xi_2]$ | Relative transmissibility of species types | Gamma$(1, 1)$ |
| $[\zeta_1, \zeta_2]$ | Relative susceptibility of species types | Gamma$(1, 1)$ |
| $\nu_0$ | Shape of seasonality | Gamma$(2, 2)$ |
| $\nu_1$ | Timing of seasonality | Beta$(2, 2)$ |
| $\omega$ | Exponent in transmission kernel | 1.3 |
| $a$ | Shape of gamma distribution for time to notification | 4 |
| $b$ | Scale of gamma distribution for time to notification | 2 |

 

cleaning and disinfection, and reduced risk of contact with wild birds and contamination of water sources, feed storage and housing, and the potential use of ring vaccination in response to IPs. We consider in our model that all these measures will have the effect of reducing the susceptibility of the poultry that could become infected with HPAI, and so the risk of HPAI incursion.

In our model simulation strategies, enhanced control will be mandated on the discovery of the presence of HPAI infection within a premises from the time of culling and will last for a specified number of days. This will either occur in all poultry premises within a particular radius (5 km, 10 km, or 15 km) of the IP or within all poultry premises in the same county or region (see Fig B in S1 Text for full details on Great British counties and regions). The effect of enhanced control is to reduce the susceptibility to infection of the nearby premises due to the reduced risk achieved by the improved control measures. In practice, this scales our $\beta_{ij}$ term (which includes the premises susceptibility component) by a given proportion, termed the susceptibility factor, while we leave the background infection term $\epsilon$ unchanged. In this manuscript, we consider enhanced control measures that reduce susceptibility to 80%, 60%, 40%, and 20% of the baseline susceptibility level that was determined in model fitting (a susceptibility factor of 0.8, 0.6, 0.4, or 0.2) for 7, 14, 21 or 28 days since the date of culling on an IP.

## Results

### Verifying the model fitting

The MCMC process was successful in providing posterior parameter distributions for each fitted parameter, with good convergence of the chains. Posterior parameter estimates are described in detail in Figs D–E and Table A in S1 Text, and we note that these estimates remained broadly unchanged when considering an exponential kernel, rather than the results for the Cauchy kernel presented here.

Sampling from the joint posterior distributions to simulate the model forward in time from the observed initial conditions on 1 October 2022, we verify that we achieve a good correspondence back to the data (Fig 2). See Fig J in S1 Text for these results presented in terms of the total number of infected poultry. The data points for the weekly number of premises reported as infected fall within the 95% prediction intervals of the model simulations, although generally towards the lower end of these model projections (Fig 2A). However, the closest-matching model trajectories show strong agreement with the data, indicating that the model can successfully replicate the outbreak and provides a good temporal fit. In these model simulations, 27.8% (95% prediction interval: 14.1%–51.6%) of the IPs arise due to the background term $\epsilon$, with the remainder due to the local infection components.

To consider the spatial model fit, we divide the number of IPs from the national model simulations into the eleven geographical regions of Great Britain. We observe that we have achieved a favourable match to the data spatially in most of these regions, with the proportion of the total IPs within the regions having a close median value in simulations to the real-world data for the season (Fig 2B). The two notable exceptions to this occur in the East of England and Scotland. For the East of England, we predict a larger proportion of cases than observed, with the inverse true of Scotland. This can, in part, be explained by the initial conditions of the model simulations, as there happen to be initially many IPs in the East of England and few in Scotland, as per the data.

Therefore, we highlight the difficulty of achieving an optimal model fit to this data set, both spatially and temporally simultaneously, due to the large state space of possible outcomes and relying on random spillover events in poultry premises to recreate the observed localised outbreaks seen in the data set. This is despite demonstrating that the model can generate simulations with comparable results to the original data. This overall quality of the model fit is shown on the map in Fig 2C. The blue dots, which represent the IPs in the data set, typically fall within the 10 km × 10 km grid cells that are predicted to be at the highest risk of infection in the model.

We additionally calculate that there will likely be few 'occult' infections at the end of our simulations, with a median number of 2 IPs identified in model fitting. The true value in the data of 0 (from data post-September 2023), therefore, falls

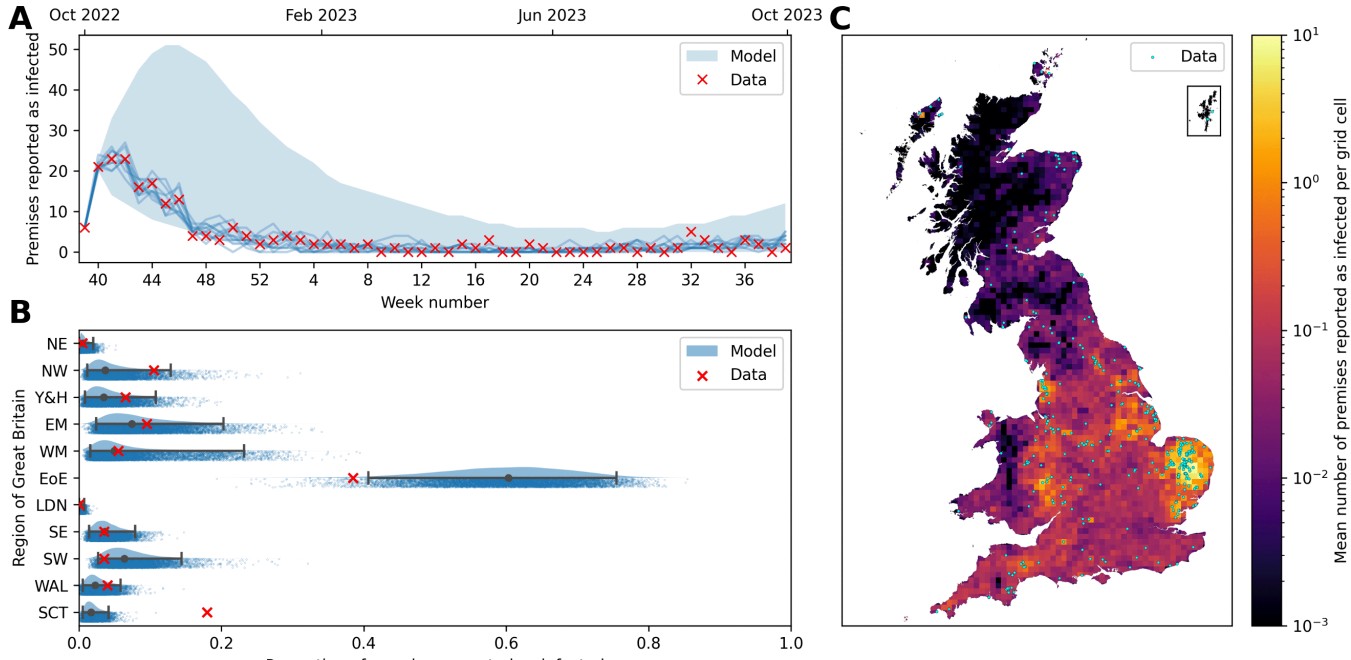

**Fig 2. Comparison of model simulations to premises notification of infection data for 1 October 2022 – 30 September 2023.** (A) Time series of the weekly number of premises with reported infections. The red crosses indicate the data, while the shaded blue regions show the 95% prediction intervals of stochastic model simulations. Blue lines indicate the best-fitting 10 individual realisations of the simulation out of the total 10,000. Note that the initial narrow model prediction intervals are due to the initial conditions of infected premises that do not become notified until the second week. (B) Raincloud plots [59] for the proportion of infected premises (IPs) across the eleven geographical regions of Great Britain. The data are represented by red crosses, while the central prediction interval and rainclouds show model simulations. The grey dot shows the median value, and the whiskers give the full 95% prediction interval. Above each interval is a half-violin plot of the distribution of the simulations, and below is a jittered scatter of each individual simulation. The names of the regions in full are: NE – North East, NW – North West, Y&H – Yorkshire and the Humber, EM – East Midlands, WM – West Midlands, EoE – East of England, LDN – London, SE – South East, SW – South West, WAL – Wales, SCT – Scotland. (C) Map of Great Britain divided into 10 km × 10 km grid cells coloured to show the mean number of IPs in each cell for the 2022–23 season. Blue dots overlaying the grid cells show the locations of the true IPs in this season. Source of map boundaries: Office for National Statistics licensed under the Open Government Licence v.3.0. The shapefiles used can be found at https://geoportal.statistics.gov.uk/datasets/5a393192a58a4e50baf87eb4d64ca828_0/explore.

within the prediction intervals. Further details on the 'occult' infections and the quality of the model fit are presented in Fig F in S1 Text.

## Control strategy scenarios

Model projections for the effect of enhanced control on the total number of IPs show that improvements may be possible by a concerted effort to reduce the potential for transmission in the vicinity of premises that have previously detected infection. If biosecurity can be improved or vaccines delivered, such that premises within an enhanced control zone that lasts for 21 days are 40% less likely to be infected (susceptibility factor = 0.6), then the median number of IPs nationally can be reduced by up to 53% (depending on the size of the enhanced control zone) (Fig 3). However, a susceptibility factor of 0.8 can still reduce the median number of IPs by up to 35% as well as a reduction in the uncertainty.

The size of the enhanced control zone has a large impact on the efficacy of the scenario, with the additional 5 km zones only causing a modest reduction in the median number of IPs (27% reduction), even when the susceptibility factor is 0.2. Increasing this radius to 10 km or 15 km impacts a much larger number of premises and so results in a larger reduction in the number of IPs, particularly when combined with lower susceptibility factors. Scaling the zone up to the full county or region shows further improvements in reducing the number of IPs. This indicates it is likely insufficient to solely

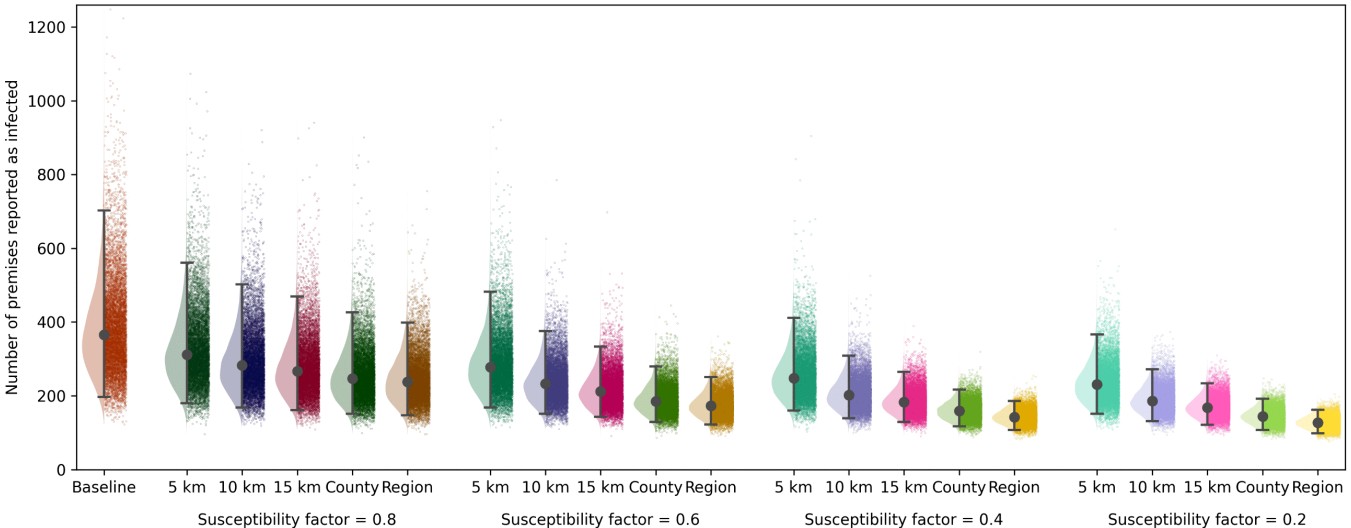

**Fig 3**. **Projected impact of enhanced control.** Each raincloud plot [59] shows the total number of infected premises (IPs) in the season for a particular enhanced control scenario, based on the region in which premises are affected (with a 5 km, 10 km, or 15 km radius, or across the county or the region) and the reduction in susceptibility due to improved control, which is here called the susceptibility factor. Estimates are obtained from 10,000 model simulations for each scenario. A susceptibility factor of 1 indicates no change in susceptibility, whereas numbers less than 1 show multiplicative factors that result in reduced susceptibility to HPAI infection. In each raincloud, it is assumed that enhanced control measures affect the given region for 21 days after poultry have been culled on the IPs, and premises within multiple impacted regions have reduced susceptibility according only to the most recent IP. Within the rainclouds, we show the median simulation value (grey dots), and 95% prediction interval (whiskers), alongside a half violin plot of the distribution (to the left) and jittered scatter of all simulations (to the right).

consider the immediate area around an IP, likely due to the movement of wild birds. Larger zones for either strict to moderate control, with increased surveillance, could improve the reduction in infection. In particular, greatly enhanced control (susceptibility factor = 0.2) combined with enacting this over the full geographical region greatly reduces the median number of IPs as well as the uncertainty in this number, eliminating some of the worst-case scenarios.

The duration of enhanced control has a relatively smaller impact on the number of IPs across the season than the susceptibility factor or size of the zone (Fig 4). There is very little difference in IPs for the season when the duration of enhanced control is varied between 7 and 28 days. This emphasises that most of the impact is due to quickly implementing the enhanced control zone upon detection of infection, with diminishing returns for keeping the zone in place for many weeks. Secondary premises will most likely become infectious close to when the original IP is detected. The largest impact of the duration occurs when the susceptibility factor is small (such that the enhanced control is having a large effect) and the zone area is also small. In this scenario, there is a large benefit in terms of susceptibility for local premises being within the zone, but local, as yet undetected IPs may fall outside this small area, and so increasing the duration of the enhanced control zone prevents more infections.

The largest uncertainties occur when the susceptibility factor is large, because then there is little impact of the additional control measures. Similarly, there is also more uncertainty when the control zone is smaller. Stringent control measures, coupled with implementing them across larger areas, limit the potential for large outbreaks to occur. We note that the lower prediction interval is not impacted substantially, because we keep the background infectious pressure term $\epsilon$ constant across all premises and only reduce the susceptibility due to the local infection component.

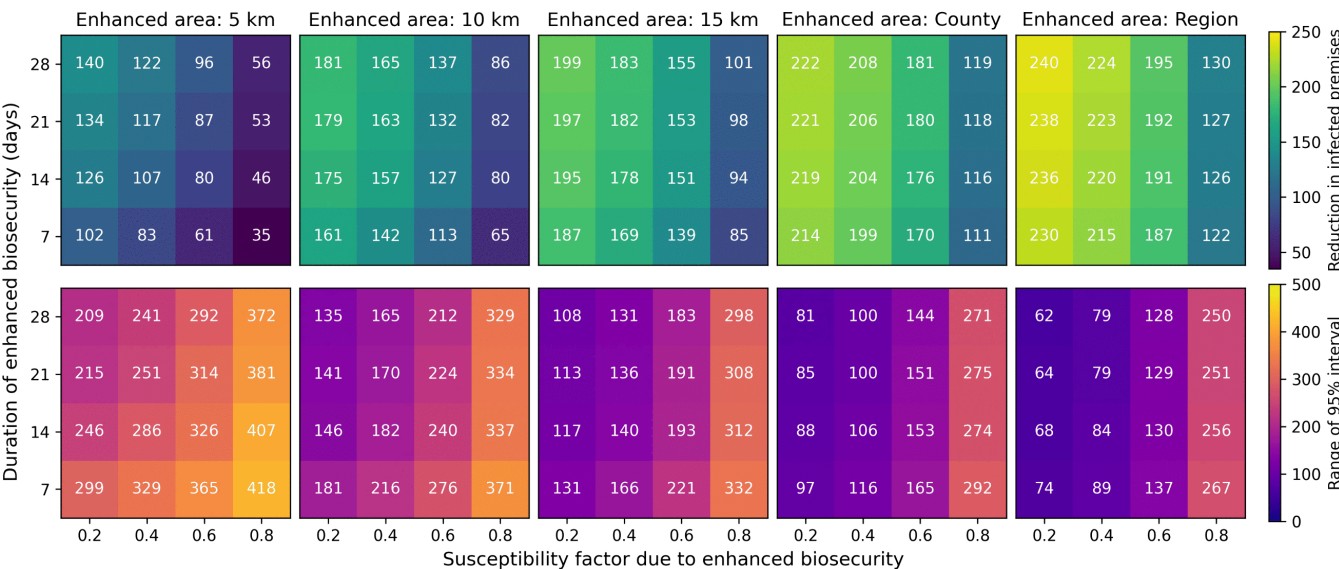

**Fig 4**. **Heat maps of the median reduction in number of infected premises (IPs) and 95% prediction interval range under different scenarios.** The top row of panels shows the median reduction in IPs for a given enhanced control duration, susceptibility factor, and zone. The bottom row shows the respective 95% prediction interval range. Estimates are obtained from 10,000 model simulations for each scenario. Each panel shows the results for different enhanced control zone areas.

## Discussion

Overall, this study showcases a spatial model for HPAI infection in the poultry of Great Britain that has been newly adapted from previous livestock modelling studies [33,45–47] to model H5N1 clade 2.3.4.4b. The model is shown to capture the observed IP data for the 2022–23 season, but could also be applied to other time periods to similarly estimate the underlying infection dynamics. The complex fitting process is computationally intensive; given our data on IPs, the inference procedure identifies distributional estimates for our model parameters. We also used the model to determine the most likely as yet undetected IPs by the end of our data period (or the 'occult' infections), such that the updated transmission process by the inclusion of these infections would most improve our model fit.

We have shown that our mechanistic modelling approach can be additionally used to perform counterfactual simulations for the season under consideration. We chose to consider enhanced control measures to limit the potential for further transmission events within a certain area by reducing the susceptibility of poultry within these areas. These simulations indicate that additional measures around premises that have had confirmed infections can greatly reduce the total number of premises that experience infections. The greatest impact occurs when enhanced control is established in the premises within a large area centred on an IP. There is a substantial reduction in the number of additional IPs, even if just the immediate premises within a 5 km radius make efforts to reduce the potential for HPAI transmission. We note that if spillover from wild birds is the primary route by which poultry become infected, biosecurity measures to minimise contact with wild birds are of critical importance.

Our results therefore align with the national policy that there is a benefit to implementing increased biosecurity in a radius around IPs [30]. However, we have highlighted that there is an additional benefit in terms of reducing the number of IPs to increasing the size of this radius (Fig 3), although clearly this will come at an additional cost. Vaccinations could

also offer a solution to reduce the susceptibility of poultry and hence reduce the number of IPs, although there are practical and commercial issues with delivering vaccinations, as well as increased costs [29]. In particular, the use of vaccinations would result in reduced mortality and fewer symptomatic HPAI infections in poultry, decreasing the probability of detection and increasing costs associated with surveillance [60].

We also note that there will be variations in how strictly control measures are applied, particularly in small-scale poultry premises. National surveys in the UK have indicated there are differences in the actions of small-scale poultry keepers in regards to both awareness and compliance with restrictions, as well as trust in authorities [11,61]. Therefore, practically, it may be difficult to uniformly implement enhanced control strictly across all the required areas.

In designing our model, we have made the simplifying assumption that the background infectious pressure from wild bird spillover into poultry premises is spatially uniform across Great Britain. This assumption could be adjusted by incorporating spatial information on wild bird habitats and detected cases into the $\epsilon$ term in the model equations (Eq 2). Known environmental sources of infection or reported wild bird cases could potentially be added as pseudo-premises to the model to include additional transmission sources. However, since reported wild bird case numbers rely on the passive surveillance of found dead birds, there will likely be substantial underreporting and biases in the locations where dead birds are more likely to be found. Therefore, incorporating these data could skew the model results.

Alternatively, using only the current data sets, future work could investigate fitting the underlying model parameters of the background infection term ($\epsilon_0$, $\nu_0$ and $\nu_1$) separately for each region of Great Britain. This could resolve issues with the spatial model fit, at the cost of additional model parameters to estimate. We could also explicitly account for the impact of biosecurity and housing orders within the model fitting process by introducing more parameters, rather than assuming these effects are captured within the baseline parameters. This could improve the model fit by enabling the model simulations to include additional spatial heterogeneity compared to our current results. However, in this manuscript, we emphasise that we have been able to achieve a remarkably good match to the real-world data for the 2022–23 season, given the lack of these spatial factors in the model (Fig 2).

In considering components of our spatial model fit, the biggest weakness is in matching the total number of IPs in the East of England and Scotland. However, this is related to the complexity of forward simulating a full year of HPAI transmission from the real-world initial conditions of 1 October 2022, alongside previously mentioned considerations. In the East of England, there were initially 23 IPs at the start of the season, and so a majority of model projections will favour continued transmission here, since there are many initially infected premises. Transmission within this region does not rely on new random introductions from wild bird spillover. In contrast, in Scotland, model simulations are initiated with a single IP (on the relatively remote Isle of Lewis) and so rely on further chance model events of spillover in this region to generate the substantial number of infections observed across the region in the data. The discrepancy is therefore not a failure of model fitting (given the spatially invariant model parameter $\epsilon$), but highlights the challenge of reproducing stochastic outputs across a long time period.

We have also assumed that the latent period and time to culling are fixed and the same for all IPs, in order to reduce the complexity of the fitting process. However, our chosen values are typical of those found in the literature [52]. We have chosen to omit continued transmission of infection in the area around an IP after culling the poultry to reduce our model complexity. Therefore, we may underestimate the benefit of continued enhanced control measures for longer periods after notification.

To further refine the model, we would like to understand the mechanisms behind transmission by untangling which transmission events come directly from wild bird spillover events versus those that originate from other infected poultry premises. This would enable us to better identify the risks associated with premises reporting infections. At present, the transmission kernel in our model includes many transmission routes: direct transmission between premises, transmission through an intermediary wild bird or birds, transmission through shared contaminated resources of poultry premises including poultry workers, and transmission directly from wild birds due to the area being a hotspot of HPAI infection, observed due to other local IPs.

Therefore, future adaptations of the model could involve considering industry links between poultry premises, using wild bird infection data, or considering phylogenetic analyses, albeit outside of the scope of the present study. In addition, if the current HPAI vaccination policy was to change to allow the vaccination of poultry, we could adapt our enhanced control strategy parameters to specifically consider vaccination, rather than only considering it as part of a package of interventions that could be used to reduce HPAI susceptibility in poultry [29]. By modelling vaccination explicitly, we could also consider the impact of reducing the transmissibility of HPAI amongst poultry that were both vaccinated and infected, which would not occur with other enhanced control measures, and would likely lead to even fewer infected IPs. Time delays due to vaccine implementation or vaccination developing protective immunity could also be incorporated.

HPAI continues to be a worldwide concern due to the devastating impact on both poultry and wild bird populations, particularly in recent years, combined with the fact that further reassortment events could lead to the emergence of a virus with the potential to cause pandemics. For the recently circulating H5N1 virus in birds of Great Britain, we have demonstrated the potential for epidemiological models to reduce key features of outbreak dynamics and highlighted the possibility for enhanced control measures to reduce the impact of HPAI outbreaks in the Great British poultry industry in the future.

## Supporting information

**S1 Text. Additional information on data and model fitting.** A description of the data used alongside details of the model assumptions and outputs from the model fitting process.
(PDF)

## Acknowledgments

The authors would like to thank Alexander Mastin and Ruth Moir from the Animal and Plant Health Agency (APHA) and Helen Roberts from the Department for Environment, Food and Rural Affairs (DEFRA) for the provision of demographic and case data used in this work.

## Author contributions

**Conceptualization:** Chris P. Jewell, Robin N. Thompson, Michael J. Tildesley.

**Data curation:** Michael J. Tildesley.

**Formal analysis:** Christopher N. Davis.

**Funding acquisition:** Edward M. Hill, Robin N. Thompson, Michael J. Tildesley.

**Investigation:** Christopher N. Davis.

**Methodology:** Christopher N. Davis, Edward M. Hill, Chris P. Jewell, Robin N. Thompson.

**Software:** Christopher N. Davis.

**Validation:** Edward M. Hill.

**Visualization:** Christopher N. Davis.

**Writing – original draft:** Christopher N. Davis.

**Writing – review & editing:** Christopher N. Davis, Edward M. Hill, Chris P. Jewell, Kristyna Rysava, Robin N. Thompson, Michael J. Tildesley.

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
