## [Decision Letter · Decision Letter 0]

9 Oct 2025

PCOMPBIOL-D-25-00966

A modelling assessment of the impact of control measures on highly pathogenic avian influenza transmission in poultry in Great Britain

PLOS Computational Biology

Dear Dr. Davis,

Thank you for submitting your manuscript to PLOS Computational Biology. After careful consideration, we feel that it has merit but does not fully meet PLOS Computational Biology's publication criteria as it currently stands. Therefore, we invite you to submit a revised version of the manuscript that addresses the points raised during the review process.

Please submit your revised manuscript within 60 days Dec 08 2025 11:59PM. If you will need more time than this to complete your revisions, please reply to this message or contact the journal office at ploscompbiol@plos.org. Please include the following items when submitting your revised manuscript:

We look forward to receiving your revised manuscript.

Kind regards,

Konstantin B. Blyuss

Academic Editor

PLOS Computational Biology

Jennifer Flegg

Section Editor

PLOS Computational Biology

**Journal Requirements:**

3) Some material included in your submission may be copyrighted. According to PLOSu2019s copyright policy, authors who use figures or other material (e.g., graphics, clipart, maps) from another author or copyright holder must demonstrate or obtain permission to publish this material under the Creative Commons Attribution 4.0 International (CC BY 4.0) License used by PLOS journals. Please closely review the details of PLOSu2019s copyright requirements here: PLOS Licenses and Copyright. If you need to request permissions from a copyright holder, you may use PLOS's Copyright Content Permission form.

Potential Copyright Issues:

i) Figures S1, and S6. Please (a) provide a direct link to the base layer of the map (i.e., the country or region border shape) and ensure this is also included in the figure legend; and (b) provide a link to the terms of use / license information for the base layer image or shapefile. We cannot publish proprietary or copyrighted maps (e.g. Google Maps, Mapquest) and the terms of use for your map base layer must be compatible with our CC BY 4.0 license.

4) Thank you for stating "Data on HPAI cases in poultry and Great British poultry premises demography are available on request from the Animal and Plant Health Agency (APHA)." For studies involving third-party data, we encourage authors to share any data specific to their analyses that they can legally distribute. PLOS recognizes, however, that authors may be using third-party data they do not have the rights to share. When third-party data cannot be publicly shared, authors must provide all information necessary for interested researchers to apply to gain access to the data. For more information, see:

https://journals.plos.org/ploscompbiol/s/data-availability#loc-acceptable-data-access-restrictions

4) All necessary contact information others would need to apply to gain access to the data.

3) If any authors received a salary from any of your funders, please state which authors and which funders.

6) Please provide a completed 'Competing Interests' statement, including any COIs declared by your co-authors. If you have no competing interests to declare, please state "The authors have declared that no competing interests exist". Otherwise please declare all competing interests beginning with the statement "I have read the journal's policy and the authors of this manuscript have the following competing interests:"

**Reviewers' comments:**

Reviewer's Responses to Questions

Reviewer #1: GENERAL COMMENTS

This study provides a mathematical, mechanistic model of highly pathogenic avian influenza (HPAI) transmission and control in Great Britain during the 2022-23 epidemic. Although adapted from previous studies, the model is fitted to the observed data on infected premises in Great Britain, which is to my knowledge the first time this has been done for this country. As such, this is one of the strengths of this study, which uses advanced fitting method to achieve this aim. The model and methodology are sound and well described, and the code and scripts are available on GitHub which is appreciated and will ease reproducibility. Beyond these methodological aspects, the authors also evaluate how further reducing susceptibility of poultry farms (e.g., through improving biosecurity measures such as cleaning and disinfection or potentially through reactive vaccination) could have impacted the 2022-23 epidemic. Results suggest that major outbreaks could have been avoided with such enhanced control measures, especially if they are implemented in large areas around IPs. This study is timely and can both be used as a reference to model HPAI in other contexts and also to inform policy. I believe it is of interest to the readership of the journal.

However, before publication, I have a few major comments that need to be addressed, and some minor comments that could contribute to improve the clarity of the manuscript.

MAJOR COMMENTS

One of my first comment is that you should make clear everywhere in the text (including abstract) that what you tested is the impact of a reduced susceptibility. You did not model explicitly vaccination or enhanced biosecurity measures, just assumed that those were potential ways of reducing farms susceptibility. You notably mention in the discussion that you could in the future explicitly model vaccination and I agree. Please highlight the differences you would bring to the model if you were to do that (see also my comment below).

You mention in the introduction possible transmission pathways of HPAI between poultry premises, either directly or indirectly. In particular, you discard airborne transmission and you mention that premises-to-premises transmission is likely due to the movement of vehicles. It should be made clearer which transmission mechanisms are captured by your force infection and which ones are not, if any. Also, specify that you do not explicitly account for transmission through vehicle movements (e.g., network model). Also, I understand that in your force of infection you cannot distinguish direct premises-to-premises transmission from spillover from wild birds as they are all intermingled, however I strongly suggest to quantify how many IPs were due to the background infection (first term of the infectious pressure) and how many were due to local infections (second and third terms of the infectious pressure). This would help better understand the results and their interpretation (see comment below).

Finally, and probably most importantly, when seeing the baseline scenario on figure 3, I am a bit worried: the 200 observed IPs in the data are at the lower end of the 95% prediction interval, which makes me wonder on the quality of the model fit and seems to be in contradiction with Fig 2A. I would expect the predicted number of reported IPs to have a distribution around the observed number of IPs, i.e., the median simulation value of the baseline scenario closer to 200… Please provide explanations for this.

MINOR COMMENTS

Abstract

For clarity (see my comment above), change “Our results indicate that enhanced biosecurity measures and/or vaccination” by “Our results indicate that reducing susceptibility (e.g., through enhanced biosecurity measures and/or vaccination)”

Author summary

“over the course of a season” -> “over the course of an epidemic season”

Introduction

Line 13: “hundreds of infected premises” -> I think you could be more precise and provide the actual number of IPs

On seasonal pattern vs endemic circulation, please explicit when seasonal patterns were observed and when endemic circulation was observed, e.g., was it during the 2022 summer?

Lines 23-26: “The majority of transmission to poultry has generally been attributed to wild ducks, geese, and gulls and in most cases is due to environmental contamination of infected faecal matter in water sources or direct contact with infected carcasses [16].” -> I am not convinced by this sentence, and I did not find supporting evidence for this in reference 16. I am especially struggling to see how direct contact with infected carcasses could represent a major transmission pathway from wild birds to domestic poultry, especially in a western European context. Although I would be more incline to consider contamination of water as a potential transmission route, I don’t know if this has been evidenced in Great Britain or other European countries? I would tend to consider that there would be some safeguards in place to ensure the sanitary status of the water distribution circuits in commercial poultry farming.

Lines 30-31: “Phylogenetic analyses have identified premises-to-premises transmission as being likely for only a few select IPs [21].” -> Please specify that this is true for 2020-2022

Lines 32-33: “Where premises-to-premises transmission does occur, it is likely due to the movement of vehicles or shared equipment between premises [22, 23].” -> Reference [23] is from the Republic of Korea. Although in agreement with the authors’ point, studies in other contexts (e.g., France) showed limited role of vehicle movements in premises-to-premises transmission. This should be discussed. Moreover, you never mention again movement of vehicles later in the text.

Lines 33-35: “Airborne transmission between premises is unlikely since evidence suggests that airborne particles containing HPAI virus can only travel very short distances (up to 10 metres) [24].” -> I would suggest to be more precise, as airborne transmission of HPAI in general IMO remains unclear and probably requires further investigation. For instance: “During the 2022-23 season, airborne transmission between premises was unlikely since evidence suggests that airborne particles containing HPAI virus only travelled very short distances (up to 10 metres) [24].”

Line 37: replace “movement” by “introduction”

Lines 48-50: Did you include changes in transmission dynamics in your model to account for these housing orders that were not in place at the beginning of the 2022-23 season?

Lines 50-52: did you account for the effect of protection and surveillance zones in your model? Were there any reactive culling involved (i.e., culling around infected premises)?

Lines 64-66 and 67-71: I think you should highlight even more in the introduction the fact that this is (at least to my knowledge) the first mechanistic model fitted to HPAI epidemic data in Great Britain and not just a simulation model, which is one of the originalities of your study.

Methods

Lines 86-87: “We had demographic data from 1 December 2022, which falls within our fitting period.” -> What do you mean “from 1 December 2022”? Do you have temporal information on your demographic data? Later on I understand that you have the data registered on 1 December 2022, is that correct?

Lines 109-110: “all susceptible poultry will be culled unless specific exemption criteria apply” -> what kind of exemption criteria are we talking about? Does this mean that some infected poultry could be left in place?

Lines 159-160: “premises, where there is an additional multiplicative scaling parameter γ1 applied to the force of infection from notified premises” -> This has already been said, I would remove it here.

Lines 160-161: “We also consider an exponential kernel in S1 Text.” -> Maybe explicit that this is to see how the shape of your kernel impacts your results.

Line 172: “but there is little specific literature on between-flock latency periods [50].” -> Although I agree, you have some more information in this other literature review (https://doi.org/10.1186/s13567-023-01219-0) on previously used between-flock latency periods that are in line with your assumed value of four days.

Lines 176-177: “This allows for individual differences dependent on the specific premises, but provides an estimate that falls within the typical distribution [50].” -> I would rephrase to avoid confusion, as you did not estimate this parameter. Maybe you could also give the mean of the distribution (in days) to help the reader get a better idea of the duration of the time to notification.

Line 184: how do you choose the value of the shape of the transmission kernel?

Text S1: “We observe that none of these parameters diverge significantly for the prior estimates, but do form smooth distributions that are unique from the priors, indicating that the fitting process has been successful (Figure S4).” -> I am not sure I understand what you mean, and the two first part of the sentence seems a bit contradictory. Were your posterior distributions informed by your data or not? Please clarify.

Lines 189-195: please better define what you are calling ‘occult’ infections. In this paragraph, you mention that these are infected but not yet notified, but you do not mention that this is at the end of your simulations (i.e., at the end of the 2022-23 period), i.e., to account for potential missing IPs in your data set at the end of the season.

On occult infections and time to notification: I do not really understand from your description how that worked. In Text S1, you mention that updating the time of notification and adding/removing occult infections are additional events that represent 5% of the total number of IPs. However, later on, you explain the results on occult infections and time to notification as if they were estimated in your model. Please provide further explanations.

Text S1: “Indeed, we truly observed zero IPs for the time scale of occult infections on 30 September 2023, in agreement with the modelled results.” -> I am not sure to understand this sentence. By definition “occult infections” are not (yet) notified and so cannot be observed, how can you be sure that there were no occult infections in real life?

Line 199: this quite a lot of infected premises at the beginning of the season. Does this mean that there was a persistence of the virus during the summer of 2022?

Lines 201-202: “The numbers were determined from our data set.” -> How? In particular, how did you defined them as exposed/infected/notified?

Line 203: Please provide here or in Text S1 a few lines explaining the grid-based system and how it works (e.g., do you model the number of susceptible/exposed/infected/notified/removed farms in each grid cell instead of modelling individual farms?). Also, specify in the main text the size of the grid cells.

Line 211: “the potential use of vaccination” -> please specify that you are talking about reactive or ring vaccination, but not preventive vaccination.

Lines 211-213: “All these measures will have the effect in the model of reducing the susceptibility of the poultry that could become infected with HPAI, and so the risk of HPAI incursion.” -> “We consider in our model that all these measures will have the effect of reducing the susceptibility of the poultry that could become infected with HPAI, and so the risk of HPAI incursion.”

Line 217: How did you apply 5-10-15 km radiuses if you used 10 km × 10 km grid cells?

Line 218: why did you not test a country-wide scenario? You mention that there were national AIPZ and housing order in place, so why not consider the possibility of national enhanced control?

Results

Lines 227-228: In addition to this sentence, please add:

- A table with some summary statistics (e.g., median or mean and 95% CrI) for the priors and posteriors of the sixteen parameters that were estimated. This would be useful in addition to Figure S4.

- One sentence saying that the posterior parameter estimates remain broadly unchanged when considering an exponential kernel and referring to Text S1.

Lines 239-244: From Figure 1 it also seems that most IPs in Scotland occurred late in the season (maybe related to infections in sea bird colonies?). Could the observed discrepancy in Scotland also come from the fact that your background infection term in the infectious pressure is temporally but not spatially variable?

Figure 2A: could you explain why you have so little variability in your simulations between week 39 and week 40? Is this related to the initial conditions?

Lines 253-256: see my comments above in the Methods on occult infections.

Lines 266 and 278-279: it seems indeed that reducing the susceptibility factor reduces the uncertainty by removing most of the worst-case scenarios and lowering the upper bound of the 95% prediction interval. In contrast, it does not seem to have a big impact on the lower bound of the interval, i.e., there are still dozens or even hundreds of IPs despite the enhanced control measures. How do you explain this? Does the reduced susceptibility impact the background infection pressure and if not, could it be an explanation?

Figure 4: the values of the duration are missing on the y-axis

Discussion

You mention the potential difficulty of having a homogenous enhanced control and it is definitely relevant. However, I am missing other potential limitations of improved biosecurity, e.g., is this technically and logistically feasible? Is it possible to further improve beyond what is already done and improved through AIPZ and protection/surveillance zones? Do you think poultry farmers still have some room for improvements? Usually, biosecurity measures are already demanding a lot from them, so I wonder how much more it is possible to ask from them.

Lines 373-377: I agree. Please discuss how considering specifically vaccination could change your results, e.g., there could be a delay before vaccination has an effect (time needed to develop protective immunity) as in reference [33], there could be an effect not only on susceptibility but also on transmissibility…

Reviewer #2: The authors modelled the transmission dynamics between poultry premises across Great Britain between October 2022 and September 2023. They developed an individual-based spatial compartmental model and fitted the demography (spatial coordinates and poultry number) and case data of poultry premises in Great Britain. Using the fitted parameters, they stochastically simulated the epidemics of the season, and projected the impacts of reducing susceptibility by controls, when varying the strength and scale (vicinity of previously infected premises) of control measures. Their main conclusions were that the model captures the temporal and spatial dynamics of the number of affected premises, and that increasing the size of control area radius and control measure strength would be beneficial. The study well showcases using modelling and data to understand the spatial dynamics of avian influenza between farms/poultry premises and control impacts, but there are a few things that need to be addressed.

Major revisions

1. Although the demography data include the number of poultry by species (authors also used this for transmissibility parameter) and I assume the case data have the number of reported cases, the model only fits the number of affected premises, without fitting case number of each premise. Therefore the major conclusions of fitted results and projected impacts are all based on the number or proportion of affected premises. It would be exciting to see if the fitted results can recover the reported cases in Figure 1, and how control measures would impact the number of cases.

2. Regarding the background infection directly caused by spillover from wild birds in the model, this parameter assumes the spillover occurs without spatial heterogeneity as the authors have addressed in Discussion, but also another assumption here is that it assumes the spillover events occur to all farms - another possibility though is that spillovers are limited and cause the initial (few) introductions in a region and the continuing onward transmission is due to transmission among premises. It would be exciting if authors could model this alternative scenario.

3. Related to the last point, the deviations of E England and Scotland simulated results from the data - is it possibly due to that the outbreaks were independent or separate from other regions? For example, a spillover event in Scotland that happens in summer (according to Reference 14) when the seasonal forcing of background infection can’t fully capture (Figure S9); and for E England, perhaps there weren’t so much background infection as the parameter represents, and were mostly local transmission between poultry premises as they start with many affected premises already. Perhaps the authors can simulate a few scenarios to test the possibilities; or, drawing data of wild bird cases to inform the parameter of background infection (spillover from wild birds).

Minor revisions

4. Line 123: “susceptible to infection” should be “susceptible to exposed”

5. Line 127: “according” should be removed

6. Figure 4: the values for duration of control (on the axis) are missing

7. Line 26-28, a suggestion on describing the species-level difference in susceptibility: “Chickens and turkeys infected with HPAI typically show more severe symptoms or higher mortality compared to ducks and geese, although the latter may have similar levels of viral shedding without showing as much symptom or mortality. This difference, however, may be less obvious for some genotypes of the circulating H5N1 clade 2.3.4.4b, as they are particularly well adapted to ducks.” The citations here can add this article of experimental infections of two genotypes of clade 2.3.4.4b of various species including chickens, ducks and geese.

Bordes L et al. 2024 Experimental infection of chickens, Pekin ducks, Eurasian wigeons and Barnacle geese with two recent highly pathogenic avian influenza H5N1 clade 2.3.4.4b viruses, Emerging Microbes & Infections, 13:1, 2399970, http://dx.doi.org/10.1080/22221751.2024.2399970

**Have the authors made all data and (if applicable) computational code underlying the findings in their manuscript fully available?**

Reviewer #1: Yes

Reviewer #2: **No:** The authors did not provide the dataset; instead, they offered availablity by request to the related government agency. If the data are not publicly available due to legal and ethical concerns, they should be stated in the data statement.

PLOS authors have the option to publish the peer review history of their article (what does this mean?). If published, this will include your full peer review and any attached files.

Reviewer #1: No

Reviewer #2: **Yes:** Qiqi Yang

**Figure resubmission:**
---

## [Decision Letter · Decision Letter 1]

1 Dec 2025

PCOMPBIOL-D-25-00966R1

A modelling assessment of the impact of control measures on highly pathogenic avian influenza transmission in poultry in Great Britain

PLOS Computational Biology

Dear Dr. Davis,

Thank you for submitting your manuscript to PLOS Computational Biology. After careful consideration, we feel that it has merit but does not yet fully meet PLOS Computational Biology's publication criteria as it currently stands. Therefore, we invite you to submit a revised version of the manuscript that addresses the points raised during the review process.

We look forward to receiving your revised manuscript.

Kind regards,

Konstantin B. Blyuss

Academic Editor

PLOS Computational Biology

Jennifer Flegg

Section Editor

PLOS Computational Biology

**Reviewers' comments:**

Reviewer's Responses to Questions

**Comments to the Authors:**

Reviewer #1: GENERAL COMMENTS

The authors made a good effort of clarification and accounted for most of both reviewers’ comments. I believe the manuscript has been improved. I only have a few minor comments. Lines below refer to the version with tracked changes.

The authors chose not to provide the number of IPs caused by the background term versus the number of IPs caused by the local terms. While I understand the reason behind their decisions, I still think that this is important for the interpretation of the alternative scenarios with reduced susceptibility. Indeed, this reduced susceptibility is only applied to the local terms (which should be made clearer lines 240-261) and not to the background terms, which explains some aspects of the results presented in Figure 3. Although I agree that this does not provide information on the contribution of wildlife as opposed to premises-to-premises transmission, I believe that this result could still be provided if worded carefully so as not to be misinterpreted by the reader. This substantial contribution of the background term should also be added in the explanation lines 340-342.

Regarding spatio-temporal changes of the interventions (e.g., housing orders), you mention that these will be captured within fitted parameter values. Although I agree, these will be captured on average over the entire study region and the entire study period. Do you think that explicitly accounting for spatio-temporal interventions could have improved the spatio-temporal fit of your model, e.g., reduce the overestimated number of outbreaks in some of your simulations? If yes, you could briefly discuss this in your manuscript.

MINOR COMMENTS

Line 51: I missed this during the first round of review, but what do you mean by “improved fencing”? I can see how fencing of pastures can be effective for domestic mammals (cows, pigs…) to reduce contacts with wildlife, but I am having trouble seeing how this can work for birds.

Line 137: maybe you could add “exposed to HPAI infection E (i.e., infected but not yet able to transmit infection)”

Line 139-141: there is a potential confusion between the time of infection, the time at which premises became exposed E, and the time at which premises became I. For clarity, maybe the I compartment should be renamed as infectious instead of infected, and make sure that there is no confusion between the time of infection (i.e., when premises became exposed) and the onset of infectiousness (i.e., when premises moved from E to I).

Line 144-145: is the background time-varying term only capturing infections caused by spillover from wild birds, or could it also implicitly capture other transmission routes, e.g., long-range transmission by vehicles movements that are not captured by the local components?

Lines 232-239: why is your addition appearing in red and crossed out? This is useful and should appear in the final manuscript.

Lines 340-342: maybe you should explain why the background term is not impacted by reduced susceptibility. I guess it is because it would apply to all farms in the country, whereas you are interested in local improvements around IPs?

Table S1: “parameter” (an "e" was missing)

Reviewer #2: The authors have well addressed my first comment and the minor comments.

Regarding my second major comment, it seems to be not completely understood and I would like to clarify. By "another possibility though is that spillovers are limited and cause the initial (few) introductions in a region and the continuing onward transmission is due to transmission among premises.", I meant that the initial introductions could be limited in some premises in a spatial region. Since the background infection is modelled as a constant in the model for all the premises, the model can't account for this spatial heterogeneity. The authors mentioned they have discussed this limitation in the article and I believe they were referring to this paragraph: "In designing our model, we have made the simplifying assumption that background infectious pressure from wild bird spillover into poultry premises is spatially uniform across Great Britain. This could be challenged by incorporating spatial information on wild bird habitats and detected cases into the ϵ term in the model equations (Equation 2). Alternatively, known environmental sources of infection or reported wild bird cases could be added as pseudo-premises to the model to include additional transmission sources. However, we have shown that we are able to achieve a remarkably good match to the real-world data for the 2022–23 season, given the lack of this information in the model (Figure 2)." However, this explanation is different from authors' response to my comment where the authors stressed the limitation is due to lack of data.

Reported wild bird cases could be accessed from Food and Agricultural Organization of the United Nations (FAO) Empres-i and World Animal Health Information System database provided by the World Organization for Animal Health. It seems that authors could incorporating thses data into ϵ or adding wild bird cases as pseudo-premises, as they suggested themselves to address the spatial heterogeneity of introduction or background infection.

If these approaches are both limited by some other issues, I suggest the authors add other limitations or edit the relevant text in the discussion to provide a more sufficient explanation. Nontheless, I believe my other suggestion of simulating senarios of different levels of background infection would still work, without modifying the model - since the model is individual premise based, the authors could have varying background infection terms for premises in different region. For example, to test if the deviation of Scotland data from the model is caused by a separate introduction in the summer and continuing local transmisison within Scotland, the authors can simulate four senarios: 1. specifiy a different ϵ for all premises in Scotland, and separate these premises from the rest (assuming there is no transmission between Scotland and all other premises); 2. only specify a different ϵ for all premises in Scotland; 3. only separate Scotland premises from the rest; 4. the originally modelled scenario, and see which result would better reflect the data. Again, if these tests would not be possibly made using the current model or data, I sugges the authors to include relevant limitations and future work in the article.

**Have the authors made all data and (if applicable) computational code underlying the findings in their manuscript fully available?**

Reviewer #1: Yes

Reviewer #2: Yes

PLOS authors have the option to publish the peer review history of their article (what does this mean?). If published, this will include your full peer review and any attached files.

Reviewer #1: No

Reviewer #2: **Yes:** Qiqi Yang

**Figure resubmission:**
---

## [Decision Letter · Decision Letter 2]

24 Dec 2025

Dear Dr. Davis,

We are pleased to inform you that your manuscript 'A modelling assessment of the impact of control measures on highly pathogenic avian influenza transmission in poultry in Great Britain' has been provisionally accepted for publication in PLOS Computational Biology.

Best regards,

Konstantin B. Blyuss

Academic Editor

PLOS Computational Biology

Jennifer Flegg

Section Editor

PLOS Computational Biology

Reviewer's Responses to Questions

**Comments to the Authors:**

Reviewer #1: The authors have addressed all my comments and I believe the manuscript is now ready for publication.

Reviewer #2: The authors have addressed my comments and I look forward to seeing the progress of the authors' on-going work of incoporating the spatial heterogeneity of the background infection.

**Have the authors made all data and (if applicable) computational code underlying the findings in their manuscript fully available?**

Reviewer #1: Yes

Reviewer #2: None

PLOS authors have the option to publish the peer review history of their article (what does this mean?). If published, this will include your full peer review and any attached files.

Reviewer #1: No

Reviewer #2: **Yes:** Qiqi Yang

---

## [Editor Report · Acceptance letter]

PCOMPBIOL-D-25-00966R2

A modelling assessment of the impact of control measures on highly pathogenic avian influenza transmission in poultry in Great Britain

Dear Dr Davis,

I am pleased to inform you that your manuscript has been formally accepted for publication in PLOS Computational Biology. Your manuscript is now with our production department and you will be notified of the publication date in due course.

With kind regards,

Anita Estes
